# AMPS: Adaptive Modality Preference Steering via Functional Entropy

## Abstract

Multimodal Large Language Models (MLLMs) often exhibit significant modality preference, which is a tendency to favor one modality over another. Prior work has applied steering methods to adjust the modality preference of MLLMs. However, these conventional approaches apply a uniform steering intensity to all samples. This lack of adaptation is problematic because strong steering can disrupt a model's standard inference capabilities, leading to high error rates, while weak steering may be ineffective. To address this limitation, a sample-wise diagnostic tool is required to measure MLLMs' susceptibility to steering across different multimodal samples. To reduce the disruption of strong steering to MLLMs' inference capabilities, we first introduce a diagnostic metric that quantifies the information contribution ratio from each modality in MLLMs. This metric reveals varying susceptibility to steering across different samples. Building on these diagnostic insights, we further propose a steering scaling strategy that applies lower steering intensity for samples highly sensitive to steering, and design a learnable steering module that automatically learns appropriate scaling patterns, enabling context-aware adjustment of modality preference. Experimental results show that our context-aware scaling method outperforms conventional steering strategies in modulating modality preference, achieving effective adjustment while significantly reducing generation errors.

## 1 Introduction

Multimodal Large Language Models (MLLMs) have demonstrated remarkable capabilities in processing and integrating different modalities such as text, images, and audio, enabling complex tasks like visual question answering, multimodal dialogue, and cross-modal reasoning (Liu et al., 2023b; Wu et al., 2025a; Li et al., 2022). One critical ability of MLLMs is to perform reasoning across various available modalities. However, the MLLMs exhibit significant modality preference and present an inherent tendency to favor a part of modalities (Li et al., 2024; Zheng et al., 2025), which occasionally does not met users' expectation in practice. For example, in visual questioning and answering combined with text context, MLLMs may disproportionately rely on text context while underutilizing image inputs. This calls for controllable methods for adjusting MLLMs' modality preferences (Li et al., 2024).

Recent studies have explored applying steering methods to adjust the modality preferences of MLLMs (Parekh et al., 2025). These methods typically involve introducing calibrated steering vectors to shift the model's activations towards the target modality. However, a common and critical limitation of these conventional steering methods is their application of a uniform steering intensity across all input samples (Zhang et al., 2025a). This uniform steering intensity strategy is fundamentally problematic because the susceptibility of a MLLMs output to steering can vary dramatically depending on the input context. Applying an excessively strong steering signal to a sample that is inherently sensitive may disrupt the model's standard inference process, leading to a high rate of generation errors and degraded performance (Zhang et al., 2025a). Conversely, for samples that are robust to changes, a weak steering signal may prove insufficient to elicit any meaningful shift in modality preference, rendering the intervention ineffective. This lack of adaptation thus poses a significant challenge to the practical deployment of steering methods. As shown in Figure 1a and 1b, we select 100 samples where Qwen2.5VL-7B have preference on either text or image and applying a mean steering to adjust modality preference to the other modality. When steering intensity

becomes sufficiently strong, the ratio of generation collapse significantly increases, limiting further preference steering.

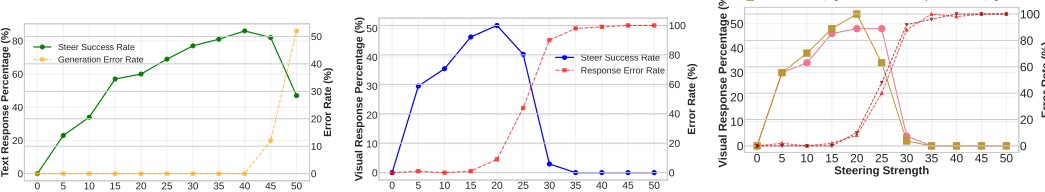

(a) Visual-to-text Meaning Steering  (b) Text-to-visual Meaning Steering  (c) Visual-to-text Meaning Steering Sample Classified with MCS

Figure 1: Mean Steering with different steering intensity.

One prospective way to address this challenge is providing an adaptive steering intensity across different sample, which necessitates a sample-wise diagnostic tool that can quantify an MLLM's susceptibility to steering; however, such a tool remains under-developed. To bridge this gap, we first introduce a novel diagnostic metric Modality Contribution Score (MCS) that quantifies the information contribution ratio from each modality within the model's reasoning process. This metric serves as a crucial probe, revealing that different samples indeed exhibit varying inherent susceptibility to steering interventions.

Building on these diagnostic insights, we propose an adaptive steering framework to achieve more precise and robust modality preference control. Specifically, we first design a steering scaling strategy that dynamically adjusts the steering intensity based on the MLLMs reliability toward different modalities as indicated by the Modality Contribution Score (MCS). We applying lower steering intensity on MLLMs when it has low reliance on the opposite of target modality to prevent generation collapse, while employing stronger steering for more robust samples with high modality reliance to ensure effective preference shift. To further automate and optimize this process, we introduce a learnable steering module adaptive modality preference steering (AMPS) that goes beyond heuristic scaling. This module is trained to automatically infer and apply appropriate, context-aware scaling patterns directly from the input, enabling a more nuanced and data-driven adjustment of steering intensity. Our approach effectively mitigates the limitations of uniform steering, paving the way for more reliable and practical deployment of steering methods in MLLMs. The main contributions of our work can be summarized as follows:

- We propose a diagnostic metric MCS to quantify modality contribution and measure MLLMs' sample-wise susceptibility to steering.
- We introduce a context-aware steering strategy that adapts steering intensity per sample.
- We develop a learnable module AMPS to learn appropriate scaling patterns of steering intensity, and validate its improved effectiveness over existing steering methods.

## 2 RELATED WORK

**Modality preference in MLLMs.**   A growing line of work argues that large multimodal models exhibit stable *modality preferences* under cross-modal conflict. Zhang et al. (2025a) introduce MC$^2$ to elicit controlled image–text inconsistencies, quantify preference directions across 18 MLLMs, and show that preference is encoded in latent representations that are steerable at inference time. Concurrent analyses report systematic lean-toward-text effects ("blind faith in text") across vision-centric tasks (Deng et al., 2025) and dataset/model-side skew measured via a Modality Importance Score (MIS) for video QA (Park et al., 2025). Beyond measurement, several works frame preference as an instance of language-prior dominance and propose decoding-time penalties or calibration to re-weight visual evidence (Zhang et al., 2024). Together, these studies motivate diagnostics that make modality reliance observable and manipulable rather than an opaque failure mode.

**Steering methods and the need for sensitivity-aware adaptation.**   Representation-level steering is an attractive post-hoc approach: by adding targeted directions in hidden space, one can increase reliance on vision or text without retraining (Zhang et al., 2025a). However, fixed-magnitude

(global) shifts are brittle when desired behavior varies by input; recent *input-dependent* schemes predict a sample-specific steering vector (L2S), reducing hallucinations and over-correction relative to static baselines (Parekh et al., 2025). Orthogonal efforts steer through causal attention interventions (CausalMM) (Zhou et al., 2024), safety-oriented adaptive controllers (AutoSteer) (Wu et al., 2025b), and training-time preference alignment such as Noise-Aware Preference Optimization (NAPO) (Zhang et al., 2025b) or modality-fair objectives (Jiang et al., 2024). A complementary line penalizes language priors during decoding to counteract text-dominance without finetuning (Zhang et al., 2024), and model-editing studies examine whether targeted edits can debias multimodal behavior (Wang et al., 2024). These trends point to a common limitation: steering intensity that is too uniform can underperform on sensitive inputs, motivating sensitivity-aware control.

**Attention distribution, visual under-utilization, and decoding-time fixes.** Mechanistic analyses probe *where* visual signals fade inside LVLMs. Mid-layer attention lenses localize the critical stages for processing image tokens and link their degradation to object hallucination (Jiang et al., 2025). At the token level, *visual attention sinks* identify irrelevant visual tokens attracting large attention mass; redistributing this mass (VAR) restores grounding without training (Kang et al., 2025). Several decoding strategies explicitly combat long-generation drift that erodes attention to image tokens, including key–value merging with image-guided logits (IKOD) (Yang et al., 2025) and image-token-attention-guided decoding (iTaD) that contrasts intermediate layers to sustain visual grounding (Xu et al., 2025). Collectively, these results suggest that steering should be conditioned on how much visual evidence the model is currently using, rather than applied uniformly.

We build on the *modality preference* evaluation and representation-steering literature (Zhang et al., 2025a; Parekh et al., 2025) and the attention-diagnostic view of visual under-utilization (Jiang et al., 2025; Kang et al., 2025; Yang et al., 2025). Our approach follows this line but emphasizes *sensitivity-aware* (input-adaptive) steering guided by preference diagnostics; for theoretical context we relate our diagnostics to information-balancing ideas from functional-entropy regularization (Gat et al., 2020).

## 3 PRELIMINARY

### 3.1 MLLM INFERENCE FORMULATION

Let $\mathbf{x} = \{\mathbf{T}, \mathbf{V}, \mathbf{P}\}$ denote the multimodal input to a MLLM, where $\mathbf{T}$: Text tokens (instructions/questions) $\mathbf{V}$: Visual tokens (encoded image patches) $\mathbf{P}$: Prompt tokens that trigger modality preference.

Given a MLLM $p_\theta(\cdot)$ parameterized by $\theta$, the visual input $\mathbf{V}$ is first encoded by a vision encoder $e(\cdot)$ such as Vision Transformer and a multimodal projector, producing visual tokens $e(\mathbf{V})$. These visual tokens are concatenated with text tokens $\mathbf{T}$ and optional prompt tokens $\mathbf{P}$, then fed into the large language model backbone $g_\pi$:

$$\left(\mathbf{h}^{(l)}\right)_{l=1}^{L} = g_\pi^{(l)}\left(e(\mathbf{V}), \mathbf{T}, \mathbf{P}]\right)_{l=1}^{L}, \tag{1}$$

where $L$ is the total number of layers, $\pi^{(l)}$ denotes the $l$-th transformer layer, and $\mathbf{h}^{(l)}$ represents the hidden states at layer $l$.

The final layer output is projected via a logit projection layer $\phi$ to produce the output distribution:

$$p_\theta(y_t \mid \mathbf{x}) = \mathrm{softmax}\left(\phi(\mathbf{h}_t^{(L)})\right) \tag{2}$$

### 3.2 MODALITY-SPECIFIC FUNCTIONAL ENTROPY OF MLLMS

Functional entropy is defined for continuous random variables. Consider a non-negative function $f(\mathbf{z})$ where $\mathbf{z} \in \mathbb{R}^d$ is a stochastic variable with probability measure $\mu$. The functional entropy is defined as:

$$\mathrm{Ent}_\mu(f) \triangleq \int_{\mathbb{R}^d} f(\mathbf{z}) \log f(\mathbf{z}) d\mu(\mathbf{z}) - \left(\int_{\mathbb{R}^d} f(\mathbf{z}) d\mu(\mathbf{z})\right) \log \left(\int_{\mathbb{R}^d} f(\mathbf{z}) d\mu(\mathbf{z})\right). \tag{3}$$

---

**Algorithm 1** Modality Contribution Metric Implementation

---

**Require:** Model $\pi$, input $\mathbf{x}$, modality token ranges $\mathcal{R}$, KV cache $KV_{\text{prev}}$, perturbation strengths $\mathcal{E}$
**Ensure:** MCS $\mathcal{S}_m$ for modalities $m \in \{\text{V}, \text{T}\}$
1: Compute baseline: $\text{logits}_x, KV_x \leftarrow \pi(\mathbf{x}, KV_{\text{prev}})$, $\mathbf{p}_x \leftarrow \text{softmax}(\text{logits}_x)$
2: **for** $\sigma \in \mathcal{E}$ **do**
3:     **for** $m \in \{\text{V}, \text{T}\}$ **do**
4:         Sample noise: $\epsilon \sim \mathcal{N}(0, \sigma_m^2 \mathbf{I})$
5:         Perturb KV cache: $KV_z \leftarrow KV_{\text{prev}}$, $KV_z[\mathcal{R}[m]] \leftarrow KV_z[\mathcal{R}[m]] + \epsilon$
6:         Forward pass: $\mathbf{p}_z \leftarrow \text{softmax}(\pi(\mathbf{x}, KV_z))$
7:         Compute CE loss: $\mathcal{L} \leftarrow \text{CE}(\mathbf{p}_x, \mathbf{p}_z)$
8:         Compute Gradient norm: $\mathcal{C}_{m,\sigma} \leftarrow \|\nabla_{KV_z} \mathcal{L}\|_2^2 / \mathcal{L}$
9:     **end for**
10: **end for**
11:
12: **return** $\mathcal{S}_m \leftarrow \mathbb{E}_{\sigma \sim \mathcal{E}}[\mathcal{S}_{m,\sigma}]$ for each $m$

---

This quantity is non-negative, i.e., $\text{Ent}_\mu(f) \geq 0$, with equality if and only if $f(\mathbf{z})$ is constant. Direct empirical estimation is challenging due to the logarithmic integral term. Instead, we employ the logarithmic Sobolev inequality for Gaussian measures, which bounds the functional entropy by the functional Fisher information:

$$\text{Ent}_\mu(f) \leq \frac{1}{2} \int_{\mathbb{R}^d} \frac{\|\nabla f(\mathbf{z})\|^2}{f(\mathbf{z})} d\mu(\mathbf{z}). \tag{4}$$

In our case We define the following function to measure the sensitivity of the MLLMs output logits in Eq 2 to Gaussian perturbations z on modality-specific hidden states as:

$$f_M = \text{CE}\left(p_\theta(\cdot \mid \mathbf{z_m}), p_\theta(\cdot \mid \mathbf{x})\right) \tag{5}$$

where $m \in \{\mathbf{T}, \mathbf{V}\}$ denotes the perturbed modality, $\mu_m$ is a Gaussian measure centered at $\mathbf{x}_m$ with variance $\sigma_m^2$, $\mathbf{z_m}$ represents the input $\mathbf{x}$ with modality $m$ related contents perturbed, and $\text{CE}(p, q)$ denotes the cross-entropy between distributions $p$ and $q$.

## 4 METHOD

### 4.1 MODALITY CONTRIBUTION METRIC IMPLEMENTATION

To measure modality specific functional entropy of MLLMs, we apply the log-Sobolev inequality shown in Eq 4 with f as Eq 5, which gives a bounding of functional entropy with functional Fisher information, indicating the information contribution from different modalities for MLLMs. To facilitate calculation, we further apply a Monte Carlo sampling to estimate the integral.

$$\text{Ent}_\mu(f_M) \leq \int \frac{\|\nabla_{\mathbf{z}_m} \text{CE}(p_\theta(\cdot \mid \mathbf{z}), p_\theta(\cdot \mid \mathbf{x}))\|^2}{\text{CE}(p_\theta(\cdot \mid \mathbf{z}), p_\theta(\cdot \mid \mathbf{x}))} d\mu_m(\mathbf{z}) \tag{6}$$

Take the Monte Carlo sampling on the right, We define this value as modality contribution score (**MCS**) as the following.

$$\mathcal{C}_m = \frac{1}{N} \sum_{j=1}^{N} \frac{\|\nabla_{\mathbf{z}_{m,j}} \text{CE}(p_\theta(\cdot \mid \mathbf{z}_j), p_\theta(\cdot \mid \mathbf{x}))\|^2}{\text{CE}(p_\theta(\cdot \mid \mathbf{z}_j), p_\theta(\cdot \mid \mathbf{x}))} \tag{7}$$

To empirically estimate the **MCS** $\mathcal{C}_m$ derived from our theoretical framework in MLLMs, we implement a Key-Value (KV) cache based perturbation gradient analysis on the MLLMs during its forward pass. Our approach quantifies the MCS of the MLLMs through the output distribution difference to infinitesimal Gaussian noise injected into the Key-Value hidden representations of a specific modality $m$.

The procedure for our measurement each single generation step is outlined in Algorithm 1 and illustrated in Figure 2. Specifically, for a given input sequence, we firstly perform a standard forward pass to obtain the original output probability distribution from the MLLMs output logits $p_\theta(\cdot|\mathbf{x})$.

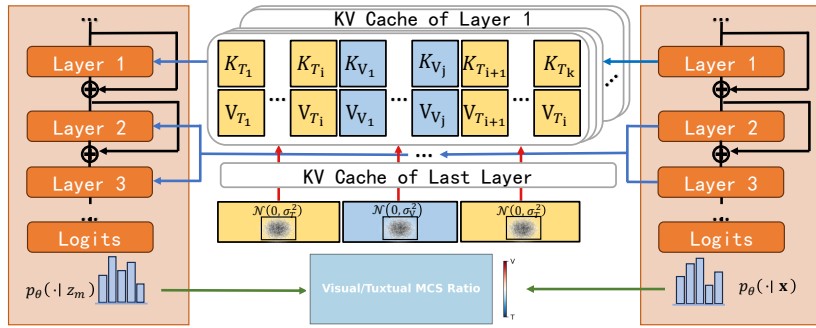

Figure 2: MCS Measurement Pipeline, in this figure the $K_{T_i}$ and $K_{V_j}$ means the corresponding to the index of the i-th text or the index of the j-th visual token in K cache; $V_{T_i}$ and $V_{V_j}$ means the corresponding to the index of the i-th text or index of the j-th visual token in V cache.

Then for each modality (Visual **V**, Text **T**), we introduce isotropic Gaussian noise $\epsilon \sim \mathcal{N}(0, \sigma^2)$ specifically to the segments of the KV cache corresponding to the target modality. A subsequent forward pass with the perturbed cache yields a new distribution $p_\theta(\cdot|\mathbf{z})$. The contribution score is calculated by computing the squared norm of the gradient of the cross-entropy loss between the original and perturbed distributions with respect to the perturbed KV cache and normalized by the loss itself. This score, averaged across $N$ perturbations, provides a robust empirical measure of the functional Fisher information and, by extension, the model's reliance on each modality.

## 4.2 Adaptive Steering Implementation

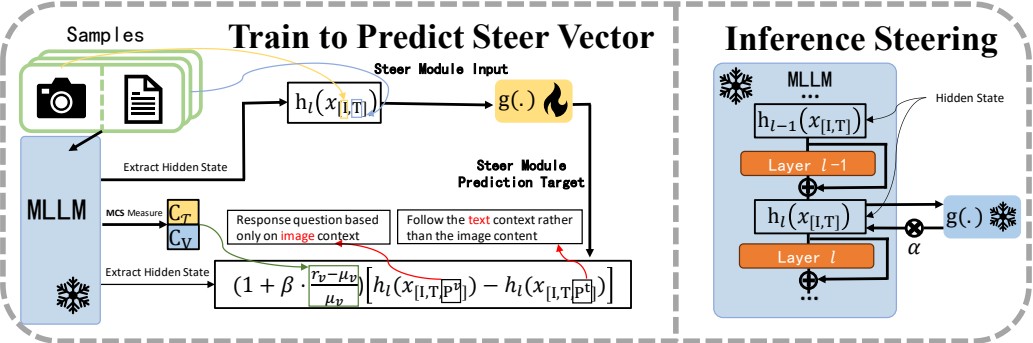

Figure 3: AMPS Training and Application Pipeline

Conventional steering methods for preference modification construct contrastive pairs by appending prompts such as $\mathcal{P}^+$ = "based on visual context" and $\mathcal{P}^-$ = "based on text context" to steer the modality bias in a desired direction:

$$X^+ = (V, T \parallel P^+), \quad X^- = (V, T \parallel P^-) \tag{8}$$

However, directly applying steering based on these steering vector extracted by such pair construction fails to provide a context-aware steering intensity scaling for preference adjustment. The key challenge is a lack of diagnostics of the severity of the model's preference deviation toward expected direction when processing each sample and adjust the intensity of steering in a fine-grained way. Previously studies () and our observation in Figure 1c both show steering uniformly with sufficient steering intensity can impair MLLMs' normal inference, which leads to failure in MLLM generation. On the other hand, insufficient steering does not impact MLLMs behaviour sufficiently. Based on our observation that MLLMs exhibit varying modality biases depending on different context, finding a context-aware scaling steering mechanism to mitigate modality bias with different intensity is a prospective solution.

**Adaptive Steering Vector Extraction.** To bridge this gap, we provide a context-aware steering mechanism, which learn to generate steering vector with different steering intensity based on the severity of MLLMs' modality preference, as shown in Figure 3. Specifically, during extraction of the steering vector, we save the visual and textual MCS of MLLMs, and then take the average of visual MCS ratio across all samples in the same task as an anchor ratio $\mu_v$, since MLLMs does not completely fail on whole tasks, we use the deviation of visual ratio on each sample $r_v$ to the $\mu_v$ of each sample to decide its severity of modality preference and use this to scale steering intensity.

$$\gamma = \left(1 + \beta \cdot \frac{r_v - \mu_v}{\mu_v}\right) \tag{9}$$

where $\mu_v$ is the anchor visual ratio, $r_v$ is the current visual ratio, $h_l$ represent the hidden state at the index of $|I, T, P_v|$ and $|I, T, P_t|$ respectively, and $\beta$ is used to adjust the context-aware intensity.

Following previous learn-to-steer setting, we extract the hidden states at a layer L of the MLLM on each constructed data pair, and subtract them to form a targeted steering vector. As for the input of learn-to-steer module, we extract the hidden states corresponding to the input image and query without any modality preference prompts.

$$h^{(l)}_{[I,T,P_v]} = g^{(l)}_\pi \left([f(V), T, P_v]\right), \quad h^{(l)}_{[I,T,P_t]} = g^{(l)}_\pi \left([f(V), T, P_t]\right), \quad h^{(l)}_{[I,T]} = g^{(l)}_\pi \left([f(V), T]\right). \tag{10}$$

Combine Eq 9 and Eq 10, we apply the scaling which form the predict target of the steering module.

$$v^{(l)}_{[I,T]} = (1 + \gamma) \cdot \left(h^{(l)}_{[I,T,P_v]} - h^{(l)}_{[I,T,P_t]}\right) \tag{11}$$

**Steering Module Training.** The goal is to train a learn-to-steer module $g_\Theta$ that predicts $v^{(l)}$ in Eq 11 from $h^{(l)}_{[I,T]}$ as the third term of Eq 10, with an optimization objective formulated as:

$$\Theta^* = \arg\min_\Theta \mathbb{E}_{[I,T]} \left\| v^{(l)}_{[I,T]} - g_\Theta \left(h^{(l)}_{[I,T]}\right) \right\|^2_2 \tag{12}$$

In practice, we follow conventional learn-to-steer to construct $g_\Theta$ with a light-weight 2-layer MLP as the auxiliary network and train the network using all the samples we collected. The details of steering module training is illustrated in the left part of Figure 3.

**Steering Module Inference.** As shown in Figure 3, we apply the light-weight trained module on one layer's hidden states of MLLMs which predict a scaled steering vector and apply the steering vector back to the hidden state of MLLMs at this layer during MLLMs inference, following conventional l2s experiment setting, we take the 14-th layer in our experiment.

$$\tilde{h}^{(l)} = h^{(l)} + \alpha \cdot g_{\Theta^*} \left(h^{(l)}\right) \tag{13}$$

where $\tilde{h}^{(l)}$ is the steered hidden state at layer $l$, $h^{(l)}$ is the original hidden state, and $\alpha$ is a scaling hyperparameter controlling the intensity of steering.

## 5 EXPERIMENT

In this section, we mainly study the following research questions.

- RQ1: Can the proposed Modality Contribution Score (MCS) effectively quantify and reveal the modality preference patterns in MLLMs?
- RQ2: Does our Adaptive Modality Preference Steering (AMPS) framework outperform existing prompt engineering and static steering methods?
- RQ3: How does the proposed scaling mechanism contribute to the steering performance?
- RQ4: How does the scaling mechanism enhance the steering dynamics and robustness?

**Experiment Setup.** Our experiments are conducted on the Modality Context Conflict dataset ($\text{MC}^2$) dataset (Zhang et al., 2025a), where we diagnoses modality preference of MLLMs and compare the modality preference steering performance of AMPS with other baselines including prompt engineering, static steering methods and conventional learn-to-steer method. We evaluate

our proposed methods on Qwen-2.5VL (Bai et al., 2023) of 3B and 7B, and LLaVA1.5 (Liu et al., 2023a) of 7B and 13B, spanning various parameter scales. Note: We applied AI tools including ChatGPT to polish our paper.

## 5.1 RQ1: CAN THE PROPOSED MODALITY CONTRIBUTION SCORE (MCS) EFFECTIVELY QUANTIFY AND REVEAL THE MODALITY PREFERENCE PATTERNS IN MLLMS?

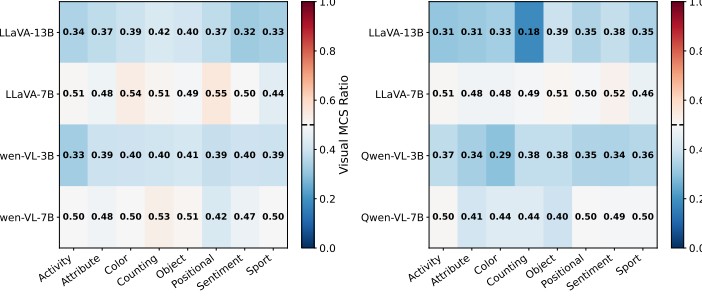
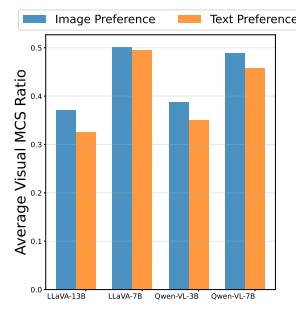

Figure 4: Sensitivity scores of different models across different tasks.

Figure 5: Visual MCS with preference prompts.

Using our proposed MCS measurement pipeline, we quantitatively diagnose the modality preference patterns of MLLMs. We compute Visual and Textual MCS values, and visualize the ratio of Visual MCS to the total MCS, on the $MC^2$ dataset.

We conduct this analysis on two model families, Qwen-VL and LLaVA, with different parameter sizes under two controlled settings designed to induce contradictory preferences, *visual preference* and *text preference*. To elicit these preferences, we append specific instructional prompts to the original query. For instance, prompts like *"Respond to the question based only on the image context"* are used to encourage visual preference, while instructions such as *"Follow the text context rather than the image content"* are used to promote text preference. The distribution of the Visual MCS ratio across different tasks is visualized in Figure 4, and the aggregated results are summarized in Figure 5. Our analysis reveals two key observations:

**MLLMs exhibit significant variation in modality reliance in different input contexts and are highly sensitive to instructional prompts.** As shown in Figure 4, the Visual MCS ratio varies considerably not only across different tasks but also among different models. This indicates that the relative contribution of visual information to the model's reasoning is not static but is highly context-dependent. Furthermore, this contribution score is systematically and predictably influenced by modality-preference prompts, demonstrating that MLLMs' intrinsic preference can be steered through external instructions.

Table 1: Performance comparison with conventional steering and preference modification methods

| Preference Model | Attribute | | | | | | | | Total |
|---|---|---|---|---|---|---|---|---|---|
| | Sport | Sentiment | Positional | Counting | Color | Activity | Object | Sentiment | |
| **Vision** | | | | | | | | | |
| MLLM-only | 26.4 | 12.4 | 0.8 | 13.2 | 4.0 | 16.0 | 11.6 | 38.0 | 15.3 |
| Inst Design | 60.8 | 24.0 | 20.0 | 20.4 | 10.8 | 32.0 | 27.2 | 63.2 | 32.3 |
| CoT Prompting | 57.6 | 27.6 | 16.0 | 18.0 | 21.6 | 35.6 | 44.4 | 52.4 | 34.2 |
| Few shot | 32.0 | 12.0 | 10.0 | 11.0 | 2.0 | 25.0 | 9.0 | 38.0 | 17.2 |
| Qwen2VL-7B | 78.8 | 35.6 | 38.8 | 29.6 | 22.4 | 56.4 | 45.2 | 78.4 | 48.1 |
| Qwen2.5VL-7B-AMPS | **67.6** | **68.0** | **85.6** | **66.8** | **88.8** | **71.2** | **52.8** | **83.2** | **73.00** |
| **Text** | | | | | | | | | |
| MLLM-only | 12.8 | 46.0 | 68.8 | 38.0 | 39.6 | 20.0 | 43.6 | 14.0 | 35.4 |
| Inst Design | 14.4 | 46.8 | 72.8 | 40.4 | 55.6 | 24.4 | 35.6 | 11.6 | 37.7 |
| CoT Prompting | 27.2 | 63.6 | 83.6 | 62.8 | 75.6 | 53.2 | 58.0 | 20.4 | 55.6 |
| Few shot | 21.0 | 77.0 | 89.0 | 73.0 | 73.0 | 60.0 | 42.0 | 42.0 | 63.1 |
| Qwen2.5VL-7B | 69.6 | 67.6 | 84.4 | 50.8 | 82.8 | 57.6 | 54.8 | 41.2 | 63.6 |
| Qwen2.5VL-7B-AMPS | **83.2** | **80.8** | **68.0** | **77.6** | **50.8** | **82.0** | **96.8** | **70.4** | **76.2** |

Table 2: Adjusting MLLMs' modality preference on $MC^2$ dataset.

| Target | Model | Task | | | | | | | | |
|---|---|---|---|---|---|---|---|---|---|---|
| | | Act. | Attr. | Color | Count. | Obj. | Pos. | Sent. | Sport | Total |
| Visual | LLaVA-7B | 12.4 | 8.0 | 12.0 | 2.8 | 33.2 | 22.8 | 0.4 | 22.0 | 14.2 |
| | LLaVA-7B-l2s | 18.0 | 11.6 | 13.6 | 6.0 | 36.0 | 16.0 | 1.6 | **25.6** | 16.05 |
| | LLaVA-7B-AMPS | **22.0** | **18.4** | **27.6** | **12.8** | **43.2** | **28.8** | **2.4** | 23.6 | **22.35** |
| | LLaVA-13B | 10.0 | 8.4 | 8.0 | 2.4 | 28.0 | 14.8 | 2.0 | 28.0 | 12.70 |
| | LLaVA-13B-l2s | 18.0 | 10.0 | 14.8 | 6.8 | 39.6 | 18.0 | 4.0 | 44.8 | 19.50 |
| | LLaVA-13B-AMPS | **47.2** | **32.8** | **54.0** | **49.2** | **66.4** | **52.4** | **16.4** | **78.8** | **49.65** |
| | Qwen-VL-3B | 28.0 | 24.0 | 41.2 | 12.0 | 60.8 | 31.6 | 6.8 | 64.0 | 33.55 |
| | Qwen-VL-3B-l2s | 30.8 | 24.4 | 55.2 | 17.2 | 68.4 | 40.0 | 6.4 | 70.4 | 39.10 |
| | Qwen-VL-3B-AMPS | **44.0** | **28.4** | **61.6** | **32.0** | **76.0** | **42.8** | **12.4** | **75.6** | **46.6** |
| | Qwen-VL-7B | 29.6 | 28.0 | 58.8 | 31.2 | 72.4 | 33.2 | 8.8 | 65.6 | 42.00 |
| | Qwen-VL-7B-l2s | 46.0 | 53.6 | 76.4 | 43.2 | 76.8 | 46.8 | 19.6 | 69.6 | 54.0 |
| | Qwen-VL-7B-AMPS | **67.6** | **68.0** | **85.6** | **66.8** | **88.8** | **71.2** | **52.8** | **83.2** | **73.00** |
| Text | LLaVA-7B | 82.0 | 91.6 | 87.2 | 87.6 | 62.8 | 86.0 | 90.8 | 74.8 | 82.85 |
| | LLaVA-7B-l2s | 78.4 | 91.2 | 85.2 | 87.6 | **78.0** | 85.6 | 90.8 | 75.6 | 84.05 |
| | LLaVA-7B-AMPS | **86.0** | **92.0** | **92.0** | **94.0** | 65.2 | **88.0** | **92.8** | **76.0** | **85.75** |
| | LLaVA-13B | 89.6 | 92.0 | 86.8 | 96.4 | 69.2 | 83.2 | 90.0 | 74.4 | 85.20 |
| | LLaVA-13B-l2s | 91.6 | 93.6 | 88.4 | **96.8** | **82.8** | 83.2 | 90.4 | **86.0** | 89.10 |
| | LLaVA-13B-AMPS | **92.8** | **95.6** | **90.0** | 96.0 | 81.6 | **85.2** | **92.0** | 82.8 | **89.50** |
| | Qwen-VL-3B | 72.0 | 76.0 | 58.8 | 88.0 | 39.2 | 68.4 | 93.2 | 36.0 | 66.45 |
| | Qwen-3B-l2s | 75.2 | 82.0 | 70.0 | 92.8 | 44.8 | 75.6 | 96.0 | 43.2 | 72.45 |
| | Qwen-3B-AMPS | **86.8** | **86.0** | **76.8** | **94.8** | **54.0** | **84.0** | **98.0** | **54.4** | **79.35** |
| | Qwen-VL-7B | 70.4 | 72.0 | 41.2 | 68.4 | 26.0 | 66.8 | 90.4 | 34.4 | 57.60 |
| | Qwen-VL-7B-l2s | 86.8 | **83.2** | 68.4 | 77.2 | **64.8** | 79.6 | **96.0** | **73.6** | 78.7 |
| | Qwen-VL-7B-AMPS | **87.2** | 82.8 | **71.6** | **83.2** | 64.0 | **83.2** | 94.6 | 73.2 | **80.2** |

*Note:* In this table l2s is traditional learn-to-steer methods, AMPS is our proposed method.

**Instructional prompts effectively shift global modality preference, as quantified by the MCS metric.** The aggregated results in Figure 5 show a clear and consistent trend: prompts designed to induce a visual preference lead to a statistically significant increase in the overall Visual MCS, while prompts inducing a text preference result in a correspondingly higher Textual MCS. This confirms that our proposed MCS metric serves as a reliable indicator of the models' internal preference dynamics.

### 5.2 RQ2: Does our Adaptive Modality Preference Steering (AMPS) framework outperform existing prompt engineering and static steering methods?

We first compare the effectiveness of our method against two baselines, prompt engineering and static prompt steering. As shown in Table 1, our method applied to Qwen2.5-VL 7B achieves a significantly larger shift in modality preference in both directions (text-to-visual and visual-to-text) compared to baseline methods. This result indicates that our adaptive approach based on learnable module exert a far stronger range of control over modality preference than either heuristic prompt design or steering methods that apply static adjustment.

### 5.3 RQ3: How does the proposed scaling mechanism contribute to the steering performance?

To study the benefit of our proposed scaling strategy, we conduct an experiment where we compare the performance of the AMPS with traditional learn-to-steer module without the scaling mechanism. As shown in Table 2, we evaluate these variants across different model architectures including Qwen2.5-VL and LLaVA1.5, and different scales from 3B to 13B. Our comparison demonstrate that while the base learn-to-steer module effectively shifts the modality preference, integrating our

scaling mechanism yields a consistent and significant performance gain. This improvement is particularly pronounced for the more challenging text-to-visual steering direction, underscoring the benefits of sample-wise adaptation.

### 5.4 RQ4: How does the scaling mechanism enhance the steering dynamics and robustness?

We further dissect how the scaling mechanism enhances steering effectiveness. Figure 6 illustrates the performance of AMPS compared to the conventional learn-to-steer module across a spectrum of steering intensities. The analysis reveals that our method achieves a more effective preference shift across a wider range of intensities while mitigating the performance collapse typically observed when excessive steering intensity is applied to sensitive samples. This demonstrates that our scaling mechanism enables a more stable and reliable steering process.

### 5.5 Ablation Study

We conduct an ablation study to validate the contribution of each component in our framework, with results presented in Table 3.

The study demonstrates that both proposed components are essential for achieving optimal performance. The learn-to-steer module alone ("w/o scaling") provides a significant boost over the baseline without steering, confirming its foundational role in modality preference adjustment. Moreover, incorporating the steering intensity scaling mechanism to form

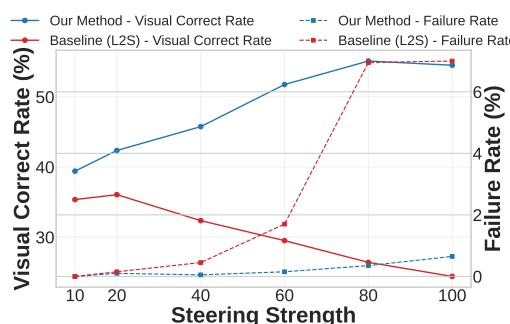

Figure 6: Comparison of AMPS and conventional learn-to-steer on different steering intensity.

the full model yields a further substantial performance gain across all tasks. This consistent improvement highlights the critical function of scaling, which adapts the steering intensity per sample, thereby maximizing effectiveness and preventing degradation.

Table 3: Ablation study on modality preference adjustment for Qwen2.5VL-3B on $MC^2$.

| Model | Task Accuracy (%) | | | | | | | | Overall |
|---|---|---|---|---|---|---|---|---|---|
| | Act. | Attr. | Color | Count. | Obj. | Pos. | Sent. | Sport | Total |
| Qwen-VL-3B (w/o steering) | 28.0 | 24.0 | 41.2 | 12.0 | 60.8 | 31.6 | 6.8 | 64.0 | 33.55 |
| Qwen-VL-3B-l2s (w/o scaling) | 30.8 | 24.4 | 55.2 | 17.2 | 68.4 | 40.0 | 6.4 | 70.4 | 39.10 |
| Qwen-VL-3B-AMPS (full) | **44.0** | **28.4** | **61.6** | **32.0** | **76.0** | **42.8** | **12.4** | **75.6** | **46.60** |

*Note:* This table presents the ablation study results. "w/o steering" denotes the baseline model without steering mechanism, "w/o scaling" indicates the model without scaling component, and "full" represents the complete proposed method. Best results are highlighted in bold.

## 6 Conclusion

In this work, we proposed a sample-wise diagnostic metric and an adaptive steering method to address the limitation brought by uniform steering intensity in MLLMs modality preference steering. We apply MCS metric to quantify the relative contribution of each modality to the model's reasoning process and revealing the varying susceptibility of different samples to steering interventions. By quantifying modality contribution and dynamically scaling steering intensity, our approach effectively adjusts modality preference while reducing generation errors. Experiments across multiple models and tasks demonstrate significant improvements over conventional steering methods. Our diagnostic analysis further confirms that MCS can reliably capture the intrinsic modality reliance of MLLMs under different input contexts, providing a principled basis for adaptive steering. Our work enables more robust and context-aware control over multimodal reasoning, with potential applications in trustworthy and interpretable MLLMs behaviour manipulation.

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
