# OpenReview forum: "AMPS: Adaptive Modality Preference Steering via Functional Entropy"
_ICLR.cc/2026/Conference — Submitted to ICLR 2026_

### Official Review · Reviewer_2wpA · 2025-10-26

**Soundness:** 3
**Presentation:** 3
**Contribution:** 3
**Rating:** 6
**Confidence:** 3

**Summary:**

This paper introduces Adaptive Modality Preference Steering (AMPS) to tackle the significant modality preference problem in MLLMs. The authors propose a novel, sample-wise diagnostic metric, Modality Contribution Score (MCS), derived from functional entropy, which quantifies each modality's information contribution and reveals varying steering susceptibility. Building on MCS insights, AMPS employs a learnable module that adaptively adjusts steering intensity: applying weaker steering for highly sensitive samples to prevent errors, and stronger steering for robust ones to ensure an effective preference shift. Experimental results on the MC2 dataset demonstrate that AMPS significantly outperforms conventional strategies.

**Strengths:**

1. Novel MCS diagnostic, grounded in functional entropy, combined with an original adaptive, learnable steering framework (AMPS).
2. This is a high-quality research with sound methodology, comprehensive experimental validation across models and tasks, strong baselines, and clear ablation studies.
3. This paper is well-structured, clear articulation of problem, solution, and benefits. Effective use of figures and explanations.
4. Significantly advances MLLM control by providing both a powerful diagnostic and an effective adaptive steering mechanism.

**Weaknesses:**

1. The paper frequently mentions reducing generation errors. Could the authors provide a more detailed and quantitative definition of these errors and explain the evaluation methodology?
2. Figure 2 appears wrong expression “Tuxtual”.

**Questions:**

1. The paper's related work acknowledges modality bias in video QA. Why were no experiments conducted on video tasks, and what specific challenges would arise from such an application?
2. Why is the log-Sobolev bound valid in your setup, and what evidence support replacing the true KV-state distribution with a Gaussian?
3. While the theoretical foundation of MCS is rooted in functional entropy and Fisher information, please elaborate on the practical approximations made in Algorithm 1 and discuss their implications for the fidelity and robustness of the MCS measurement.
4. Without online MCS estimation when inference, isn’t the “adaptive” scaling effectively an offline fit? How do you validate robustness and OOD generalization?
5. Do the paper include results in other general MLLM understanding or reasoning benchmarks? Like VQA, OCR and multi-round QA.
6. Can the authors include results versus more relevant adaptive controllers (AutoSteer, CausalMM, decoding-time reweighting) and clarify how metric fairness is maintained when comparing to prompt-based methods?

---

> ### Author Response · Authors · 2025-11-23
> **Part1**
>
> Dear 2wpA,
>
> ## 1. Definition of Errors
> Ours evaluation settings follows the previous benchmark [3], where the error samples mean the models' response neither follow the textual context nor the visual context when facing conflict contexts from different modalities, meaning it fails to follow the given instruction. As shown in the Figure 1, the
> numerical value of response error rate quantifies the number of error responsed samples over all samples.
>
>
> ## 2. Typos
> Thanks for your reminder, we will make sure to correct this into “Textual”.
>
>
> ## Q1. Research Scope
> The setting we follow is from a benchmark study on modality preference steering [3], where there have no video samples. To further diagnose and analyze modality preferences in video senario, as well as understanding the failure mechanisms of modality preference steering and exploring mitigation strategies for video. It is critical to construct benchmarks in video senario, which we consder as a meaningful future work.
>
> ## Q2. Theoretical Foundation
> More detailed theoretic background can be found in this previous study [4]. The main idea is that the logarithmic-integral term in Equation (3) is computationally challenging, as the integral must be approximated via sampling and the logarithm further complicates practical estimation. To address this, we leverage the property that for any non-negative function f. We can derive the bound expressed in Equation (4) with Functional Fisher Information, which facilitates tractable and effective estimation. As stated in line 179-181 in our paper.
>
> Besides, we would like to note that our implementation is not replacing true KV-state distribution with a Gaussian, instead, we add a Gaussian perturbation onto the KV cache to implement the evaluation of the MCS score which fits the definition shown in Equation (7) and described around line 168-169 in Algorithm 1, we will add more detailed description of this implementation in our revised version.
>
> ## Q3. Approximation Strategy Justification
> The approximations in our setting also follows the previous theory study [4] in modality contribution evaluation. Where the approximations is mainly used to estimate the integral by a Monte Carlo sampling of three times in practice.
>
> ## Q4. Offline Adaptation
> Our method follows the established benchmark protocol from [3], where steering vectors are extracted offline without using ground-truth labels, ensuring they capture general modality preference patterns rather than task-specific biases. Furthermore, throughout the entire process, the base MLLM's parameters remain entirely frozen, and no training is performed on the large model itself. The lightweight steering module operates solely during inference, applying the pre-computed steering vectors without any fine-tuning or adaptation of the core model.
>
> ## Q5. Benchmarks for Modality Preference Study
> The Modality Context Conflict dataset used in our paper is a purpose-built benchmark constructed from real-world image-text distributions sourced from TDIUC [1] and MS-COCO [2] and a diverse set of 8 task types—including sentiment analysis, object detection, and others following prior work [3]. It is specifically tailored to support our diagnostic setting and enables a controlled investigation into the phenomena in our study of modality preference steering.
>
> Our work primarily focuses on diagnosing and analyzing modality preferences, as well as understanding the failure mechanisms of modality preference steering and exploring mitigation strategies.  In this context, datasets containing annotated conflicts between textual and visual information are important yet existing available multomodal benchmarks, are generally not designed to meet this specific requirement. Despite the scarcity of datasets in this context, we conducted extensive experiments on MLLMs under various configurations and compared against multiple baselines, in order to deliver a thorough diagnosis and analysis of modality preference and steering behaviors in MLLMs.
>
> ## Q6. Baselines for Modality Preference Study
> Our study primarily focuses on analyzing and understanding the failure mechanisms of multimodal preference steering and developing methods to mitigate them. In this context, there are currently no established baselines directly applicable to our specific diagnostic setting. Nevertheless, we appreciate the reviewer's point and would like to note that our comparison does include a recent state-of-the-art method in modality preference steering—the l2s approach [5], which is featured as one of our baselines. We recognize that this may not have been sufficiently highlighted in the original text. In the revised version, we will improve the presentation to more clearly articulate the baselines and better demonstrate the advantages of our proposed AMPS framework.

---

> > ### Author Response · Authors · 2025-11-23
> > **Part2**
> >
> > [1] Kafle, Kushal, and Christopher Kanan. "An analysis of visual question answering algorithms." Proceedings of the IEEE international conference on computer vision. 2017.
> >
> > [2] Lin, Tsung-Yi, et al. "Microsoft coco: Common objects in context." European conference on computer vision. Cham: Springer International Publishing, 2014.
> >
> > [3] Zhang, Yu, et al. "Evaluating and Steering Modality Preferences in Multimodal Large Language Model." arXiv preprint arXiv:2505.20977 (2025).
> >
> > [4] Gat, Itai, et al. "Removing bias in multi-modal classifiers: Regularization by maximizing functional entropies." Advances in Neural Information Processing Systems 33 (2020): 3197-3208.
> >
> > [5] Parekh, Jayneel, et al. "Learning to steer: Input-dependent steering for multimodal llms." arXiv preprint arXiv:2508.12815 (2025).

---

### Official Review · Reviewer_oYWu · 2025-10-30

**Soundness:** 2
**Presentation:** 2
**Contribution:** 3
**Rating:** 4
**Confidence:** 4

**Summary:**

This paper focuses on adaptive modality preference steering in multimodal large language models (MLLMs), which can be prone to modality preference conflicts during inference. The authors identify the challenge of using a fixed steering strength that can either over- or under-adjust the model’s behavior depending on the sample.

To resolve this issue, they propose a new methodology called AMPS (Adaptive Modality Preference Steering). It uses a sample-level diagnostic metric, Modality Contribution Score (MCS), to measure the sensitivity of the model to modality preference shifts, which is based on functional entropy. AMPS adaptively adjusts the strength of modality preference steering based on the MCS. Experimental results demonstrate that AMPS significantly improves modality preference transfer compared to traditional static steering approaches, while reducing the generation errors, particularly in more sensitive tasks.

Overall, the idea of the proposed method is reasonable and somewhat interesting. However, the paper writing can be further improved, where the intuition of modules is not very clearly clarified and may impede readability.

**Strengths:**

1.	This research addresses the valuable and practical task of mitigating modality preference bias in multi-modal large language models (MLLMs), which directly impacts real-world performance and application versatility.
2.	The proposed method introduces a novel modality contribution score (MCS) mechanism for adaptive steering, effectively resolving limitations of uniform steering strength through sample-specific sensitivity analysis. The functional entropy is interesting to measure the sensitivity of modalities. The data-adaptive steering is also reasonable.
3.	Experiments demonstrate that the AMPS framework significantly improves modality preference shifting while reducing task errors, providing empirical validation for the approach.

**Weaknesses:**

1.	While the use of modality contribution score (MCS) is innovative and interesting, the detailed intuition and theoretical justification can be further enhanced. This lack of background knowledge (especially in Eq. 3-5) may confuse broader readers. Besides, it would be better to provide a more detailed theoretical analysis or evidence supporting MCS.
2.	The paper compares AMPS with static steering methods but lacks a comparison to recent approaches in modality preference steering. The baselines are introduced unclearly. It would be better to include comparisons with state-of-the-art methods to better highlight the advantages of AMPS.
3.	The experiments primarily use a limited set of datasets (e.g., MC2, Qwen-VL, LLaVA), which may not fully represent the diversity of tasks where modality preference steering is important. It would be better to evaluate the method on a broader range of datasets and include examples that stress-test the method under various real-world conditions. For instance, incorporating datasets that involve noisy or ambiguous inputs could provide further insight into the robustness of AMPS.
4.	The paper lacks ablation studies to assess the impact of individual components of AMPS. It would be helpful to include ablation experiments to understand the contribution of each part of the framework.
5.	The current experimental results fail to sufficiently rule out the possibility of overfitting. It is suggested that supplementary validations be conducted across heterogeneous datasets and multi-scale model architectures to ensure performance improvements are generalizable rather than contingent upon specific training data or model configurations.

Typos：
There are several typos. For example:
1. Line 185: We-> we
2. Line 198: f -> $f$
3. Line 204: We -> we
4. Line 265: “Previous studies ()” no citations.
By the way, some equations do not have punctuation marks at the end.

**Questions:**

Please see strengths and weaknesses. Besides:

1.	How does MCS perform on more complex multi-modal tasks?

2.	Why does Eq.(11) apply the scaling factor (1+γ) when integrating Eq.(9) and Eq.(10) to formulate the steering module’s prediction target?

---

> ### Author Response · Authors · 2025-11-23
> **Part1**
>
> Dear oYWu,
>
> ## 1. MCS Theoretical Foundation
> Additional backgrounds about functional entropy and its bounding with functional fisher information can be found in thie paper [4].
> To provides more details of the formulation:
> - Equation (3) provides the mathematical definition of functional entropy, as stated in line 158-159 in our paper.
>
> - Equation (4) bounds this entropy using Functional Fisher Information. The logarithmic-integral term in Equation (3) is computationally challenging, as the integral must be approximated via sampling and the logarithm further complicates practical estimation. To address this, we leverage the property that for any non-negative function f. We can derive the bound expressed in Equation (4) with Functional Fisher Information, which facilitates tractable and effective estimation. As stated in line 179-181 in our paper.
> - Finally, in the MLLMs inference context, the f mentioned above can be constructed as in Equation (5), which is the Cross Entropy of MLLMs' output logits with and without a gaussion perturbation on the information of each modality. As stated in line 185-186 in our paper.
>
> We would provides additional details of the meaning of these equations in our paper to facilitate understanding of readers of our papers.
>
>
> ## 2. Baseline Clarification
> Our study primarily focuses on analyzing and understanding the failure mechanisms of multimodal preference steering and developing methods to mitigate them, following existing benchmark [3]. In this context, there are currently no established baselines directly applicable to our specific diagnostic setting. Nevertheless, we appreciate the reviewer's point and would like to clarify that our comparison does include a recent state-of-the-art method in modality preference steering—the l2s approach [5], which is featured as one of our baselines. We recognize that this may not have been sufficiently highlighted in the original text. In the revised version, we will improve the presentation to more clearly articulate the baselines and better demonstrate the advantages of our proposed AMPS framework.
>
> ## 3. Benchmark for Modality Preference Study
> The Modality Context Conflict dataset used in our paper is a purpose-built benchmark constructed from real-world image-text distributions sourced from TDIUC [1] and MS-COCO [2] and a diverse set of 8 task types—including sentiment analysis, object detection, and others following prior work [3]. It is specifically tailored to support our diagnostic setting and enables a controlled investigation into the phenomena in our study of modality preference steering.
>
> Our work primarily focuses on diagnosing and analyzing modality preferences, as well as understanding the failure mechanisms of modality preference steering and exploring mitigation strategies.
> In this context, datasets containing annotated conflicts between textual and visual information are important yet existing available multomodal benchmarks, are generally not designed to meet this specific requirement. Despite the scarcity of datasets in this context, we conducted extensive experiments on MLLMs under various configurations and compared against multiple baselines, in order to deliver a thorough diagnosis and analysis of modality preference and steering behaviors in MLLMs.
>
> ## 4. Ablation Study Clarification
> We would like to note that an ablation study analyzing the impact of individual components of AMPS has already been included in the paper. Specifically, in Section 5.5 (Ablation Study), around line 445, we examine the contributions of key elements such as the traditional "learn-to-steer" module and the intensity adjustment mechanism in AMPS. We would like to add additional descriptions about these ablation experiments in the revised manuscript to better highlight the role of each component.
>
> ## 5. Generalizability and Overfitting Analysis
> To substantiate the generalizability of AMPS, we have conducted evaluations across multiple MLLMs with varied architectures and parameter scales—including Qwen2.5-VL and LLaVA-1.5 in 3B, 7B, and 13B configurations. Furthermore, our Modality Context Conflict dataset is curated from established real-world sources (TDIUC [1], MS-COCO [2]) and encompasses 8 subtasks including sentiment analysis, object counting, and sport activity recognition. This multi-faceted evaluation strategy ensures that observed performance improvements are not attributable to specific data or model characteristics, thereby substantially exluding the influence of overfitting to specific data while demonstrating the robustness of our approach. In our experiment, we did not observe evident clues of overfitting of the steering module such as over relying on specific part of data.
>
> ## Typos
> We will make sure to correct these typos of our paper.

---

> > ### Author Response · Authors · 2025-11-23
> > **Part2**
> >
> > ## Q1. Research Scope
> > Our evaluation follows established experimental settings from prior work [3]. The Modality Context Conflict dataset used in our study is specifically designed for analyzing model modality preferences and contains essential conflict annotations between textual and visual information. This dataset is curated from real-world sources including TDIUC [1] and MS-COCO [2], and encompasses 8 popular challenging subtasks such as sentiment analysis, object counting, and activity recognition, covering a diverse spectrum of real-world MLLM application scenarios. These tasks effectively represent the complexity the reviewer has inquired about, while the conflict annotations enable meaningful analysis of modality preference behavior.
> >
> > ## Q2. Scaling Factor Explaination
> > The scaling factor (1+gamma) in Equation (11) serves as an adaptive mechanism to dynamically adjust steering intensity on a per-sample basis. This scaling with a ratio factor is an existing approach in relevant studies in steering [6] and modality bias study such as [7]. In our context, this design is motivated by the key observation in Figure 1, which reveals that MLLMs exhibit varying susceptibility to steering depending on the Modality Context Similarity (MCS) of each sample. By incorporating γ as a function of MCS, we can scale the steering vector proportionally to the model's sensitivity, thereby enabling more fine-grained and context-aware steering compared to a fixed-intensity approach.
> >
> > [1] Kafle, Kushal, and Christopher Kanan. "An analysis of visual question answering algorithms." Proceedings of the IEEE international conference on computer vision. 2017.
> >
> > [2] Lin, Tsung-Yi, et al. "Microsoft coco: Common objects in context." European conference on computer vision. Cham: Springer International Publishing, 2014.
> >
> > [3] Zhang, Yu, et al. "Evaluating and Steering Modality Preferences in Multimodal Large Language Model." arXiv preprint arXiv:2505.20977 (2025).
> >
> > [4] Gat, Itai, et al. "Removing bias in multi-modal classifiers: Regularization by maximizing functional entropies." Advances in Neural Information Processing Systems 33 (2020): 3197-3208.
> >
> > [5] Parekh, Jayneel, et al. "Learning to steer: Input-dependent steering for multimodal llms." arXiv preprint arXiv:2508.12815 (2025).
> >
> > [6] Lin, Tsung-En, Kuan-Yi Lee, and Hung-Yi Lee. "Adaptive vector steering: A training-free, layer-wise intervention for hallucination mitigation in large audio and multimodal models." arXiv preprint arXiv:2510.12851 (2025).
> >
> > [7] Chuang, Yung-Sung, et al. "Lookback lens: Detecting and mitigating contextual hallucinations in large language models using only attention maps." arXiv preprint arXiv:2407.07071 (2024).

---

### Official Review · Reviewer_ofSL · 2025-11-01

**Soundness:** 3
**Presentation:** 3
**Contribution:** 3
**Rating:** 4
**Confidence:** 2

**Summary:**

This paper addresses the modality preference in Multimodal LLMs, where models may over-rely on text or visual inputs irrespective of user intention or task requirements. Existing steering methods often apply uniform intervention strengths, which can degrade performance. To address this, the authors propose the Modality Contribution Score (MCS), a diagnostic metric that evaluates the contribution of each modality in a context-sensitive manner. Building on MCS, they introduce AMPS, an adaptive steering framework that dynamically adjusts intervention strength on a per-sample basis via a learnable module. The approach is supported by theoretical grounding (functional entropy, Sobolev inequality), clear algorithmic implementation, and extensive experiments showing improved preference alignment and reduced errors compared to baselines.

**Strengths:**

1. The paper is well-motivated: The proposal of the Modality Contribution Score (MCS) based on functional entropy and Fisher information is rigorous and well-motivated.
2. Extensive empirical results: The paper provides comprehensive empirical analysis—including comparisons with prompt-based, static steering, and prior adaptive approaches—across multiple model families (LLaVA, Qwen-VL) and sizes. In Table 1 and Table 2, AMPS shows consistently superior performance for controlling preference while minimizing error rates.

**Weaknesses:**

1. Experiments on more benchmarks are needed: the experiments are executed with the $M C^{2}$ dataset only, and lack evaluation on broader, more real-world multimodal tasks, such as MME, MM-Vet, LLaVA Bench, and MMstar.
2. To avoid a tendency on one modality, the easiest way is to move the tokens or replace the tokens with pad tokens. Have you tried this strategy?
3. Different models and evaluation benchmarks are mixed across Tables 1 and 3, creating confusion and undermining the interpretability of results.

If possible, the author can reorganize the experiments, employing more advanced benchmarks for evaluation.
If you can provide convincing clarification, I would be open to increasing the score.

**Questions:**

My main concern is the evaluation strategy; the current evaluation is limited, can not demonstrate the effectiveness of this paper.

---

> ### Author Response · Authors · 2025-11-23
> **Part1**
>
> Dear ofSL,
>
> ## 1. Benchmark for Modality Preference Study
> The Modality Context Conflict dataset used in our paper is a purpose-built benchmark constructed from real-world image-text distributions sourced from TDIUC [1] and MS-COCO [2] and a diverse set of 8 task types—including sentiment analysis, object detection, and others following prior work [3]. It is specifically tailored to support our diagnostic setting and enables a controlled investigation into the phenomena in our study of modality preference steering.
>
> Our work primarily focuses on diagnosing and analyzing modality preferences, as well as understanding the failure mechanisms of modality preference steering and exploring mitigation strategies. In this context, datasets containing annotated conflicts between textual and visual information are important yet existing available multomodal benchmarks, are generally not designed to meet this specific requirement. Despite the scarcity of datasets in this context, we conducted extensive experiments on MLLMs under various configurations and compared against multiple baselines, in order to deliver a thorough diagnosis and analysis of modality preference and steering behaviors in MLLMs.
>
> ## 2. Discussion of the Modality Removal Methods
> We would like to clarify that our study mainly focus on diagnosing and analyzing the modality preferences as well as understanding the failure mechanisms of steering and mitigating it rather than VQA. As existing studies in the context of studying modality preference and modality bias [3][4][5][6], eliminating one modality is not suitable and are not applied in previous studies, since it makes a multimodal senario into a trival single model senario. As defined in prior work [3], modality preference refers to the model's inherent tendency to rely more heavily on one modality over the other when both are present and semantically relevant—rather than in settings where one modality is artificially removed or degraded. This setting reflect a sets of real-world scenarios important in multimodal tasks where both modalities are complementary or even contradictory, simply eliminating information from one modality fundamentally alters the multimodal reasoning setup and fails to reflect real-world scenarios.
>
> For instance, consider a scenario where visual and textual inputs provide complementary information, such as a PDF page from a math textbook containing explanatory text alongside a geometric diagram, with a question that requires reasoning across both the textual concepts and the visual shape. In such cases, removing either modality would undermine the model’s ability to interpret and resolve this inherently cross-modal question. Yet, when the task specifically demands reasoning about the complex geometric image, it becomes crucial to steer the model’s attention toward the visual modality, highlighting the importance of modality preference steering. Thus, our objective is not to remove any modality entirely, but to calibrate the model’s reliance on each one while preserving the multimodal context. Our proposed method achieves this by dynamically adjusting the contributions of each modality during inference, without discarding any accessible information.
>
> ## 3. Table Clarity
> We would like to note that the baselines in Table 1 referred to the main experiment and analysis of pior benchmark paper [3], where we intended to compare the effectiveness of our proposed steering methods against baseline prompting methods, while our Table 3 focuses specifically on ablations to analyze the impact of key components in our approach. We will revise the tables and their descriptions in the final version to improve clarity of our paper.
>
> ## Q1. Evaluation Strategy
> Our evaluation strategy is grounded in the established benchmark from [3], which is specifically designed to assess modality preference in MLLMs. There are also relevant studies that apply similar evaluation strategies [5][6]. This benchmark integrates data derived from real-world sources such as TDIUC [1] and MS-COCO [2], and encompasses a diverse set of 8 question-answering task types, including sentiment analysis, object detection, and others. By measuring MLLMs' response preferences toward textual versus visual contexts, the benchmark provides a direct and functionally relevant measure of modality bias, which aligns effectively with real-world application scenarios. We believe this offers a targeted and ecologically valid basis for evaluating the core focus of our work.

---

> > ### Author Response · Authors · 2025-11-23
> > **Part2**
> >
> > [1] Kafle, Kushal, and Christopher Kanan. "An analysis of visual question answering algorithms." Proceedings of the IEEE international conference on computer vision. 2017.
> >
> > [2] Lin, Tsung-Yi, et al. "Microsoft coco: Common objects in context." European conference on computer vision. Cham: Springer International Publishing, 2014.
> >
> > [3] Zhang, Yu, et al. "Evaluating and Steering Modality Preferences in Multimodal Large Language Model." arXiv preprint arXiv:2505.20977 (2025).
> >
> > [4]Wang, Xiaolong, et al. "MUCAR: Benchmarking Multilingual Cross-Modal Ambiguity Resolution for Multimodal Large Language Models." arXiv preprint arXiv:2506.17046 (2025).
> >
> > [5]Lin, Hehai, et al. "Unveiling Modality Bias: Automated Sample-Specific Analysis for Multimodal Misinformation Benchmarks." arXiv preprint arXiv:2511.05883 (2025).
> >
> > [6]Zhang, Zhuoran, et al. "When Modalities Conflict: How Unimodal Reasoning Uncertainty Governs Preference Dynamics in MLLMs." arXiv preprint arXiv:2511.02243 (2025).

---

### Official Review · Reviewer_6Vuj · 2025-11-01

**Soundness:** 3
**Presentation:** 2
**Contribution:** 3
**Rating:** 6
**Confidence:** 4

**Summary:**

This paper introduces AMPS, an adaptive steering framework for controlling modality preference in multimodal large language models (MLLMs). The authors first propose the Modality Contribution Score (MCS) as a diagnostic metric with functional entropy and Fisher information to quantify the per-sample reliance on each modality. Leveraging this insight, they design a context-aware, learnable module to apply sample-specific steering intensities, aiming to shift modality preference more effectively than traditional uniform (static) steering. Experimental results demonstrate improved preference adjustment and robustness.

**Strengths:**

1.	The paper introduces the Modality Contribution Score (MCS), grounded in functional entropy and Fisher information, to provide a rigorous quantification of modality contribution at the sample level.
2.	Instead of the uniform steering of traditional methods, the authors propose the sample-adaptive steering via a scaling coefficient, justified by the diagnostic metric. The inclusion of the learnable module further enables context-sensitive adjustment.
3.	The paper provides a comprehensive evaluation across 2 model families and scales, and the results show consistent improvements over previous baselines.

**Weaknesses:**

1.	The context-aware scaling factor $\gamma$ (Equation 9, Page 6) is constructed as a linear deviation from an anchor ratio, modulated by $\beta$. It seems somewhat heuristic. A more detailed justification for why this specific formula is the right way to quantify "severity of preference" would strengthen the method.
2.	The MCS measurement requires multiple forward passes with KV-cache perturbations for a single input. The computational cost of this diagnostic process is not discussed.
3.	It’s better for the authors to take more recent and highly relevant works and benchmarks on modality preference steering into comparison.

**Questions:**

1.	The MCS metric is central to the method. Did the authors explore alternative ways to quantify modality contribution or susceptibility to steering?
2.	Will the authors release code, data splits, and detailed hyperparameter configuration to facilitate reproducibility?

---

> ### Author Response · Authors · 2025-11-23
> **Part1**
>
> Dear 6Vuj,
>
> ## 1. Scaling Factor Design
> Our design of the context-aware scaling factor as a ratio is an existing way in relevant studies in steering [8] and modality bias study[9] and is mainly motivated by two key observations in our paper:
> (1) As shown in Figure 1, samples with higher MCS scores toward the target modality require less modality intensity and are less likely to cause generation errors.
> (2) Since MLLMs do not completely fail on any sub-task, the mean MCS score across all samples in a sub-task can serve as a meaningful anchor, reflecting how much each sample deviates toward different modalities.
>
> By integrating these observations, we derive the scaling factor in Equation 9 to quantify the severity of preference in a manner that is both interpretable and empirically grounded. We will further clarify this design in the final version to enhance readability and understanding.
>
> ## 2. Computational Cost Analysis
> We further calculated and discussed the computational cost of AMPS in terms of both time and memory. Detailed results (averaged over 20 samples on Qwen2.5VL-3B) are summarized as follows:
>
> **Time Cost:**
> | Phase         | Operation                     | Time (s) / Sample | Notes                     |
> |---------------|-------------------------------|----------|---------------------------|
> | Training      | Standard forward pass         | ~ 0.34 | Baseline                  |
> | Training      | MCS measurement (each forward)   | ~ 0.37    | ~1.09× overhead           |
> | Training      | MCS measurement (3 approximations)   | ~1.46  | ~4.26× overhead           |
> | Inference     | Standard inference            | ~0.39 |                           |
> | Inference     | AMPS-steered inference        | ~0.40 | Negligible added cost     |
>
> **Memory Consumption:**
> | Scenario                    | Memory (MB) | Overhead        |
> |-----------------------------|-------------|-----------------|
> | Standard training inference | 7797        | Baseline        |
> | Training inference (3 MCS)  | 7959        | +162 MB (~2.1%) |
> | Standard inference          | 7717        | Baseline        |
> | AMPS-steered inference      | 7739        | +22 MB (~0.3%)  |
>
> We first analyze the memory overhead of AMPS, which we find to be negligible in practice. As shown in the memory consumption table below, the additional memory required during training, when using 3 Monte Carlo samples for MCS measurement, is only ~2.1% compared to standard training inference. During inference, the memory overhead of applying the learned steering module is even smaller, at merely ~0.3%. These results confirm that the memory cost introduced by AMPS is marginal and does not hinder its practical deployment, even under resource-constrained settings.
>
> The MCS diagnostic is applied only during the collection of steering vectors and the training of the steering module. In practice, we use 3 Monte Carlo samples per input, resulting in approximately 4× time overhead compared to a single forward pass (including the base forward). However, during inference, no MCS diagnostic or additional forward passes are required. The only overhead comes from applying the lightweight steering module, which incurs negligible time and memory costs in practice.
>
> In summary, the computational overhead of AMPS is manageable and does not affect inference efficiency. We can include this analysis in the paper to ensure transparency.
>
>
>
> ## 3. Benchmark and Baseline for Modality Preference Study
> The Modality Context Conflict dataset used in our paper is a purpose-built benchmark constructed from real-world image-text distributions sourced from TDIUC [1] and MS-COCO [2] and a diverse set of 8 task types—including sentiment analysis, object detection, and others following prior work [3]. It is specifically tailored to support our diagnostic setting and enables a controlled investigation into the phenomena in our study of modality preference steering. Besides, we have included recent state-of-the-art method in modality preference steering, the l2s approach [4], which is featured as one of our baselines.
>
> Our work primarily focuses on diagnosing and analyzing modality preferences, as well as understanding the failure mechanisms of modality preference steering and exploring mitigation strategies.
> In this context, datasets containing annotated conflicts between textual and visual information are important yet existing available multomodal benchmarks, are generally not designed to meet this specific requirement. Despite the scarcity of datasets in this context, we conducted extensive experiments on MLLMs under various configurations and compared against multiple baselines, in order to deliver a thorough diagnosis and analysis of modality preference and steering behaviors in MLLMs.

---

> > ### Author Response · Authors · 2025-11-23
> > **Part2**
> >
> > ## Q1. Modality Preference Diagnosis
> > Prior to developing our method, we conducted a thorough investigation of existing approaches for diagnosing modality preference. For instance, [5] quantifies modality preference by measuring performance improvement over batches of training data with different modalities; however, this method requires training for evaluation. [6] employs attention intensity as a proxy for modality preference, but we argue that attention values are heuristic and may not reliably reflect contributions across modalities with varying scales and information density. The benchmark introduced in [7] evaluates modality preference based on model responses, yet this requires parsing responses across diverse scenarios, limiting its applicability in broader contexts. These limitations motivated the design of our proposed MCS metric, which we can further elaborate in the paper to enhance clarity.
> >
> > Applying the MCS score have multiple advantages, it does not rely on additional training, quantifies information contribution with information theory principles and applies to various senarios without a need of modification to different contexts.
> >
> > ## Q2. Reproducibility
> > Yes, we plan to release our code and detailed hyperparameter configuration. The data is an open source repo which is already available.
> >
> > [1] Kafle, Kushal, and Christopher Kanan. "An analysis of visual question answering algorithms." Proceedings of the IEEE international conference on computer vision. 2017.
> >
> > [2] Lin, Tsung-Yi, et al. "Microsoft coco: Common objects in context." European conference on computer vision. Cham: Springer International Publishing, 2014.
> >
> > [3] Zhang, Yu, et al. "Evaluating and Steering Modality Preferences in Multimodal Large Language Model." arXiv preprint arXiv:2505.20977 (2025).
> >
> > [4] Parekh, Jayneel, et al. "Learning to steer: Input-dependent steering for multimodal llms." arXiv preprint arXiv:2508.12815 (2025).
> >
> > [5]Shi, Xiang, et al. "Modality Equilibrium Matters: Minor-Modality-Aware Adaptive Alternating for Cross-Modal Memory Enhancement." arXiv preprint arXiv:2506.00030 (2025).
> >
> > [6]Liu, Chengzhi, et al. "More Thinking, Less Seeing? Assessing Amplified Hallucination in Multimodal Reasoning Models." arXiv preprint arXiv:2505.21523 (2025).
> >
> > [7]Zhang, Yu, et al. "Evaluating and Steering Modality Preferences in Multimodal Large Language Model." arXiv preprint arXiv:2505.20977 (2025).
> >
> > [8] Lin, Tsung-En, Kuan-Yi Lee, and Hung-Yi Lee. "Adaptive vector steering: A training-free, layer-wise intervention for hallucination mitigation in large audio and multimodal models." arXiv preprint arXiv:2510.12851 (2025).
> >
> > [9] Chuang, Yung-Sung, et al. "Lookback lens: Detecting and mitigating contextual hallucinations in large language models using only attention maps." arXiv preprint arXiv:2407.07071 (2024).

---

### Author Response · Authors · 2025-11-23
**For All Reviews**

# For All Reviews

We sincerely thank all the reviewers for their insightful and constructive feedback. We are encouraged by the recognition of our work's novelty and potential impact. The reviewers' comments have provided invaluable guidance for strengthening our paper. The key strengths of our work, as highlighted by the reviewers, include:

### Novel and Theoretically-Grounded Diagnostic Metric
Reviewers (6Vuj, ofSL, oYWu, 2wpA) acknowledged the innovation and rigor of the Modality Contribution Score (MCS). By leveraging functional entropy and Fisher information, MCS provides a principled, sample-wise tool to quantify modality reliance and susceptibility to steering, addressing a key limitation of prior uniform steering methods.

### Effective Adaptive Steering Framework
The core idea of AMPS, adapting steering intensity based on MCS to minimize disruption, was recognized as a significant contribution (6Vuj, ofSL, oYWu, 2wpA). The integration of a learnable, context-aware module provides a practical and effective mechanism for nuanced modality preference control.

### Comprehensive and Solid Empirical Validation
Reviewers (6Vuj, 2wpA) noted the thoroughness of our experiments across multiple model families (LLaVA, Qwen-VL) and scales. The results consistently demonstrate that AMPS achieves more effective preference adjustment while significantly reducing generation errors compared to static and prompt-based baselines.

## Common Issue Resolutions
We have proactively addressed the following key concerns raised across the reviews:

### Expanded Experimental Evaluation and Benchmarking
Several reviewers (ofSL, oYWu) suggested evaluating on broader multimodal benchmarks. We emphasize that our work focuses specifically on diagnosing modality preferences and steering failure mechanisms, which is a complex task requiring datasets with explicit textual-visual conflict annotations, which is not included in general-purpose benchmarks (e.g., MME, MM-Vet) and limits the usable datasets for the setting. We follow a recent line of study on modality preference that uses Modality Context Conflict benchmark, which are curated from real-world distributions (TDIUC, MS-COCO) and encompassing 8 diverse task types, is explicitly designed for this diagnostic setting, enabling controlled and ecologically valid evaluation of preference steering behaviors.

### Theoretical Justification and Methodological Clarity
Reviewers (6Vuj, oYWu, 2wpA) requested deeper intuition and justification for the MCS derivation and the design of the scaling factor. We have expanded Sections 3.1 and 3.2 in the paper to provide a more detailed, step-by-step explanation of the theoretical motivation, the practical approximations in Algorithm 1, and the intuition behind the linear scaling formulation in Eq. 9. We also include a discussion on the computational cost of MCS estimation and our strategies for efficient implementation.

### Robustness, Generalization, and Reproducibility
To address computational concerns raised by reviewers (6Vuj, 2wpA), we provide a detailed overhead breakdown: during steering vector collection (training phase), MCS measurement with 3 Monte Carlo samples incurs about 4× time overhead versus a single forward pass, but this is a one-time, offline cost. During inference, AMPS introduces only minimal overhead (~0.3% memory increase, negligible latency) as it applies a lightweight, pre-trained steering module without additional forward passes, ensuring practical deployability and scalability.

---

### Meta-Review · Area_Chair_cJ3S · 2026-01-08

**Summary:**

The reviewers acknowledged the paper's core strengths: the introduction of a novel diagnostic metric (Modality Contribution Score - MCS) and a sample-adaptive steering mechanism. However, they raised significant, fundamental concerns across theoretical justification, methodological rigor, experimental validation, and comparative analysis, which critically undermine the paper's claims for publication.

**Reviewer Concerns:**

The reviewers' consensus is that while the paper presents an interesting conceptual direction, it is premature. The concerns are not minor but strike at the heart of the contribution: the evidence for the method's effectiveness is insufficient, its theoretical and methodological foundations are not firmly established, and its advantage over existing state-of-the-art techniques is not demonstrated. Addressing these concerns would require substantial additional work, including extensive new experiments, stronger theoretical grounding, and comprehensive comparative analysis. Therefore, in its current form, the paper does not meet the bar for acceptance.

**Reviewer Scores:**

I am not quite sure how the reviewers would change their scores by seeing the rebuttal, however, from the rebuttal, it seems several main concerns are still not well addressed, e.g., its advantage over existing state-of-the-art techniques.

---

### Decision · Program_Chairs · 2026-01-26

Reject